# Quantitative Changes in White Blood Cells: Correlation with the Hallmarks of Polycystic Ovary Syndrome

**DOI:** 10.3390/medicina58040535

**Published:** 2022-04-12

**Authors:** Abdulrahman H. Almaeen, Abdulrahman Abdulwahab Alduraywish, Mudasar Nabi, Naveed Nazir Shah, Rahiman Shaik, Bilal Ahmad Tantry

**Affiliations:** 1Department of Pathology, College of Medicine, Jouf University, Sakaka 70214, Saudi Arabia; 2Department of Internal Medicine, College of Medicine, Jouf University, Sakaka 70214, Saudi Arabia; dr-aaad@ju.edu.sa; 3Department Biochemistry, Kashmir University, Srinagar 190006, India; mudasarnabi@gmail.com; 4Department of Respiratory Medicine Chest Diseases Hospital, Government Medical College Srinagar, Srinagar 190001, India; naveednazirshah@yahoo.com; 5Department of Biochemistry, Ex-Faculty, Jouf University, Sakaka 70214, Saudi Arabia; rahimhi@gmail.com; 6Department of Microbiology, Government Medical College Srinagar, Srinagar 190001, India; btantrey@gmail.com

**Keywords:** polycystic ovary syndrome, neutrophil count, neutrophil–lymphocyte ratio, low-grade inflammation, reproductive age women, hormonal profile, metabolic profile, metabolic syndrome

## Abstract

*Background and Objectives:* In women of reproductive age, leukocytosis is a risk factor that bridges low-grade chronic inflammation (metabolic inflammation), metabolic changes, and polycystic ovary syndrome (PCOS) and is a potential early predictor of PCOS. This study aims to explore the predictive role of quantitative changes in white blood cells (WBCs) and neutrophils in PCOS-associated metabolic changes. *Materials and Methods:* A total number of 176 blood samples were obtained from age-matched women of the reproductive period, comprising 88 PCOS cases and 88 healthy controls. Hematological, metabolic, and anthropometric indices and ultrasonic assessment were recorded. *Results:* Elevated levels of luteinizing hormone, testosterone, and lipid parameters except HDL-C levels, and the prevalence of metabolic syndrome in PCOS were statistically significant (*p* < 0.001). The neutrophil count and neutrophil–lymphocyte ratio (NLR) in PCOS patients were significantly higher (*p* < 0.001) than their counterparts. The predictive ability of the neutrophil count and neutrophil–lymphocyte ratio (NLR) for PCOS, and possibly its associating subclinical inflammation at optimum cut-off values for the neutrophil count and NLR of >46.62% (sensitivity 94.32% and specificity 74.42%) and >1.23 (sensitivity 71.59% and specificity 100%), respectively. With regard to the areas under the curve (AUC) and Youden indices, they constituted 0.922 and 0.697 for neutrophil count and 0.926 and 0.716 for NLR, respectively. The comparative ROC z-statistic value was 2.222 and a *p* = 0.026. The multiple linear regression analysis revealed no significant influence for hormonal and metabolic independent variables on the neutrophil count in PCOS cases, but, as can be expected, revealed a significant negative relationship with the other components of WBCs. *Conclusion:* In conclusion, relative neutrophilia and elevated NLR are potential cost-effective, sensitive, and specific predictors of PCOS that may also shed light on the mechanism of chronic low-grade inflammation that is characteristic of the disease.

## 1. Introduction 

Polycystic ovary syndrome (PCOS) is a common endocrinopathy affecting about 6–10% of young women at their reproductive age [1]. Per Rotterdam criteria and after exclusion of other etiologies, PCOS is defined by any two of three criteria; oligo/anovulation, hyperandrogenism, or polycystic ovaries on ultrasound [2]. Anovulatory infertility associated with menstrual irregularities is most common in PCOS women as compared to non-PCOS women [3]. Subclinical inflammation is one of the important facets of PCOS, with and without obesity, which manifests by elevated counts of white blood cells (WBCs), interleukin (IL)-6, and C-reactive protein (CRP) levels. The persistent and untreated subclinical chronic inflammation in PCOS likely plays a role in the development of insulin resistance (IR) and atherosclerosis [4,5,6]. 

The increased WBCs count in PCOS may correlate the chronic low-grade inflammation, independent of obesity, that further the metabolic and cardiovascular complications with such a syndrome [5,7,8,9]. Increased and activated WBCs release inflammatory mediators, such as neutrophilic myeloperoxidase and NADPH oxidase, resulting in the accumulation of oxygen-free radicals. This prooxidant microenvironment oxidizes and deforms LDL, which along with cellular membrane lipid peroxidation, initiates the cascade towards atherosclerosis, hypertension, and metabolic syndrome [10]. At the endothelial site of injury, neutrophils secrete the further damaging neutral endopeptidase and elastase [10,11,12]. Increased levels of WBCs, the neutrophil–lymphocyte ratio (NLR), and the platelet–lymphocyte ratio (PLR) were acknowledged as cost-effective chronic low-grade inflammatory biomarkers in PCOS [5,13,14]. 

In this study, we aim to evaluate the association between changes in WBC components vs. metabolic and hormonal biomarkers in reproductive age women with PCOS as compared to age- and BMI-matched healthy control women. The predictive value of changes in neutrophil counts and NLR for PCOS, as an explanation of the associating low-grade inflammation, was also assessed.

## 2. Patients and Methods

### 2.1. Design, Setting, and Subjects 

A one-year observational case-control study was carried out during March 2017 to March 2018. A total number of 176 women (age range of 18–35 years) of reproductive age were consecutively enrolled in this study. They comprised 88 women diagnosed with PCOS and 88 healthy fertile women. Prior to conducting this study, all protocols employed were approved by the Government Medical College Ethical Committee (Ref. No: 96/ETH/GMC/ICMR, Dated 28 October 2017), and the participants signed written informed consents. All experiments were performed in accordance with the relevant guidelines and regulations. 

PCOS women included in this study were diagnosed based on Rotterdam diagnostic criteria, where the patient should fulfill two of the three criteria: (i) Biochemical hyperandrogenism (total serum testosterone ≥ 2.5 nmol/L) or presence of hirsutism, (ii) volume of polycystic ovaries > 10 mL on ultrasound scan, and, (iii) oligo/anovulation [2,15]. Metabolic syndrome (MetS) was diagnosed, as per the National Cholesterol Education Program/Adult Treatment Panel III [16] if three or more of the following conditions were present: waist circumference ≥ 88 cm, blood pressure ≥ 130/85 mmHg; fasting serum triglycerides ≥ 150 mg/dL; high density lipoprotein (HDL) cholesterol (HDL-C) < 50 mg/dL; glucose ≥ 110 mg/dL or subjects were receiving treatment for these conditions. The inclusion criteria for the control subjects in this study was the absence of hyperandrogenism, normal ovaries on ultrasound scan, negative family history for PCOS and regular menstrual cycles (21–35 days). Cases were recruited using non-probability convenient sampling (i.e., all the available volunteering cases which fulfill the Rotterdam diagnostic criteria were included in the study). The age- and BMI-matching volunteering controls were taken from the same ethnic groups residing in the same area and selected using non-probability convenient sampling, as well. The exclusion criteria for both cases and controls in this study included: the presence of androgen-secreting tumors, hyperprolactinemia, congenital adrenal hyperplasia (17-hydroxyprogesterone-based), and Cushings syndrome, diabetes, pregnancy or lactation, smoking, age <18 years as well as >35 years, and 6 months past history of undergoing any hormonal therapy and lipid lowering drugs [17].

The collected de-identified demographic and medical history data included: age, menstrual cycle, marital status, family history, and medication. Signs of hyperandrogenism were evaluated by a physical examination. Anthropometric measurements (BMI; waist circumference) were performed according to the World Health Organization (WHO), the International Obesity Task Force (IOTF), and the International Association for the Study of Obesity criteria (IASO) [18]. Waist circumference was measured to assess the abdominal obesity for women: <80 cm is normal and ≥80 cm is central obesity. BMI was calculated as weight (kg)/height (m^2^).

### 2.2. Laboratory Investigations 

An overnight fasting blood sample from each participant was collected in a red-top tube during the 2nd to the 5th day (early follicular phase) of the menstrual cycle. The serum was recovered by centrifugation within an hour of collection and was stored at –80 °C until further analysis. Another blood sample was collected in EDTA tubes for hematological analysis and was analyzed within 2 h after blood collection using the Swelab Alfa Plus Standard Analyzer (Stockholm, Sweden). The manual differential counting of 200 WBCs on Wright–Giemsa-stained blood smears was carried out by a trained technician. The serum total cholesterol (cat# 10019), triglyceride (cat# 10725), HDL-C (cat# 10284), low-density lipoprotein cholesterol (LDL-C; cat# 10294), total triiodothyronine (cat# 54010), total thyroxine (cat# 54020), thyroid stimulating hormone (TSH; cat# 54030), follicle stimulating hormone (FSH; cat# 53020), luteinizing hormone (LH; cat# 53010), prolactin (cat# 53030), testosterone (cat# 55010) were measured using specific immunoassay kits from Human (Germany) and AU5200 autoanalyzer (Olympus, Tokyo, Japan), as instructed.

### 2.3. Ultrasonography 

All women in the study underwent abdominal ultrasound examination (SONOACE R7, SAMSUNG EDISON Co. Ltd., Seoul, Korea) in the same-day visit for the bilateral ovarian morphologic assessment. 

### 2.4. Statistical Analysis

The statistical analysis was performed using the Statistical Package for the Social Sciences Software version 20.0 (SPSS Inc., USA). The quantitative data are expressed as mean ± SD. The normal distribution was examined by the Kolmogorov–Smirnov test. The demographic and laboratory characteristics of POCS cases and healthy controls were compared using the Student’s *t*-test. The strength of association between neutrophil count and other parameters was assessed by Pearson’s correlation coefficient. A multiple linear regression analysis was performed to explore the effect of the independent variables on neutrophil counts. The ability of neutrophil count to predict POCS was evaluated by the receiver operating characteristic (ROC) area under the curve (AUC) and Youden’s index (sensitivity + specificity − 1). A Box and Whiskers chart was used to represent the data. All values were calculated and reported at a 95% confidence interval (CI) and a *p*-value < 0.05 was considered statistically significant. 

## 3. Results 

The comparative baseline demographic, hematological, and metabolic findings of PCOS and control women are shown in Table 1. The BMI was significantly higher among PCOS cases compared to the control group (*p* < 0.001). With regard to hematological parameters, neutrophil count, NLR, and platelet distribution width coefficient (PDWC) were significantly elevated (*p* < 0.001) in PCOS as compared to control women. The platelet count differed non-significantly between the two groups. Serum lipid profiling demonstrated significantly elevated total cholesterol, triglycerides, and LDL-C levels (*p* < 0.001), and significantly lowered HDL-C (*p* < 0.001) in PCOS cases as compared to controls. Hormonally, LH and testosterone levels in PCOS were higher than in controls (*p* < 0.001). PCOS women had larger mean waist circumferences (*p* < 0.001). However, we did not observe significant differences in the thyroid profile between the two groups. The morphologic assessment of right and left ovary volumes by U/S revealed larger volumes in the cases as opposed to the normal group (*p* < 0.001). Hirsutism was observed in all studied PCOS women. The prevalence of MetS among both groups is represented in Figure 1.

Figure 2 demonstrates ROC curve analysis of the neutrophil count for the prediction of PCOS, as a potential marker of low grade chronic inflammation. The maximum Youden index was estimated to be 0.697, while AUC at the 95% confidence interval was 0.922 with an optimal cut-off value of >46.62% (sensitivity: 94.32% and specificity: 74.42%) and a significant *p* value (<0.05). Furthermore, Figure 3 exhibits the pairwise ROC analysis of NLR along with neutrophil count. The AUC, Youden index, and cut-off value of NLR were 0.926, 0.716 and >1.23, respectively (sensitivity 71.59% and specificity 100%). The comparative ROC z-statistic value was 2.222, which was statistically significant (*p* = 0.026).

Table 2 presents Pearson’s correlation analysis between neutrophil count and the other independent laboratory variables amongst PCOS cases. A positive correlation between the neutrophil and lymphocyte counts, and the neutrophil count and platelet–lymphocyte ratio (PLR), was discernible (*p* <0.001).

The multiple linear regression analysis revealed the influence of the independent variables on the predictive value of the neutrophil count, as a potential marker of low-grade chronic inflammation in PCOS (Table 3). As a proportion of the total count, the WBCs subtypes (i.e., lymphocytes, monocytes, and eosinophils) exerted a negative effect on neutrophil count, whereas, the metabolic and hormonal variables lacked a significant relationship with the neutrophil count.

## 4. Discussion

PCOS is a complex infertility heterogeneous endocrinopathy associated with low-grade inflammatory, adiposity and metabolic syndrome [7,8,9]. There is increasingly evidence that inflammation plays an important role in the development of cardiovascular events, and is also linked to diabetes [19,20,21]. Several inflammatory biomarkers were employed to predict the low-grade inflammation in PCOS, such as the C-reactive protein [22], leucocyte count [5], and NLR [14,23,24]. In our study, the increased neutrophil count in PCOS was found to be statistically significant, which may explain, at least in part, the mechanism and extent of the inherent chronic low-grade inflammation. In fact, the low-grade inflammation in PCOS correlates with the accompanied insulin resistance and cardiovascular events [7,8]. Several studies reported the association between metabolic syndrome and changes in total and differential WBCs counts as a mechanism and biomarker for inflammation [25,26,27]. The increased WBCs in our study are in keeping with earlier studies, which tried to highlight their pathophysiological role in PCOS [4,7,24]. 

Increased waist circumference, total cholesterol, LDL-C, triglycerides, and decreased levels of HDL-C are potential risk factors for the development of metabolic syndrome and cardiovascular events [28,29,30]. The prevalence of dyslipidemia was found to be significantly prevailing in our PCOS women in comparison with healthy controls, which is supported by earlier studies [29,30,31,32]. The etiology for lipoprotein alteration in the PCOS group implicates an altered hormonal profile, particularly for hyperandrogenism and the correlating insulin resistance, peroxidative change, alterations in adipocytokine profile, and specific single nucleotide polymorphisms [33,34,35,36,37,38,39]. In our study, the hormonal profile of PCOS women revealed significantly elevated levels of FSH, LH, testosterone, and T4 as compared to controls, a finding strongly consistent with previous studies [31,40,41,42]. 

The multiple disturbances at endocrine, metabolic, and cellular parameters highly indicate that PCOS is a heterogeneous low-grade inflammatory-metabolic-endocrinopathy syndrome [30]. The increases in lymphocyte count observed in our PCOS women were reportedly reasoned to the stress-induced hypothalamic–pituitary–adrenal axis and cortisol level [43,44]. Francesco Orio et al. reported that leukocytosis is a potential predictor of low-grade inflammation in PCOS [5]. Some studies reported that the increased levels of NLR and platelets–lymphocyte ratio (PLR) are also associated with chronic inflammation in PCOS [13,14,25]. However, the changes in neutrophil count have not been analyzed solely as inflammatory biomarkers in PCOS. The ROC analysis of our data demonstrated that the optimal cutoff point for neutrophil percentage to significantly predict PCOS is > 46.62%, presenting sensitivity of 94.32% and specificity of 74.42%, with an AUC and Youden index of 0.922 and 0.697, respectively. These results emphasize that changes in neutrophil count alone may act as a potential low-grade inflammatory biomarker in PCOS. Similarly, our study also demonstrated that NLR is also an effective biomarker, which is in consistency with other studies [14,23,24,44,45,46].

Limitation-wise, our study has a relatively small sample size and was non-probability conveniently enrolled; despite our tight age range, stringent exclusion criteria, and homogenous ethnicity. Although the transvaginal scan is superior to transabdominal US assessment of Polycystic ovaries, ethical and cultural constraints preclude that. A gold-standard inflammatory biomarker was not employed for comparison.

## 5. Conclusions

In our study, all of the metabolic and anthropometric hallmarks of PCOS, including the prevalence of the metabolic syndrome, were higher in our cases than in healthy controls. Changes in the neutrophil count and NLR are potential predictors for PCOS and may mechanistically explain, at least partially, the associating low-grade inflammation in PCOS women. With the fact that correlation does imply causation in mind, the changes in neutrophil count and NLR could be utilized as sensitive, specific, simple, and cost-effective predictors of PCOS, and the accompanying low-grade inflammation—after validation vs. gold-standard inflammatory markers.

## Figures and Tables

**Figure 1 medicina-58-00535-f001:**
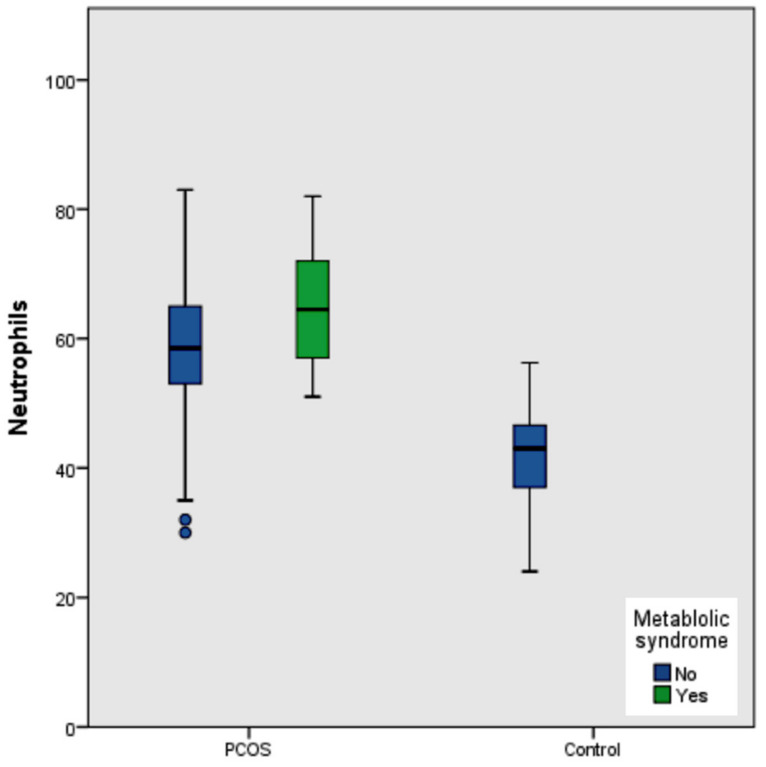
The prevalence of metabolic syndrome among the polycystic ovary syndrome cases vs. healthy control women and its association with changes in neutrophil counts. PCOS; Polycystic Ovary Syndrome.

**Figure 2 medicina-58-00535-f002:**
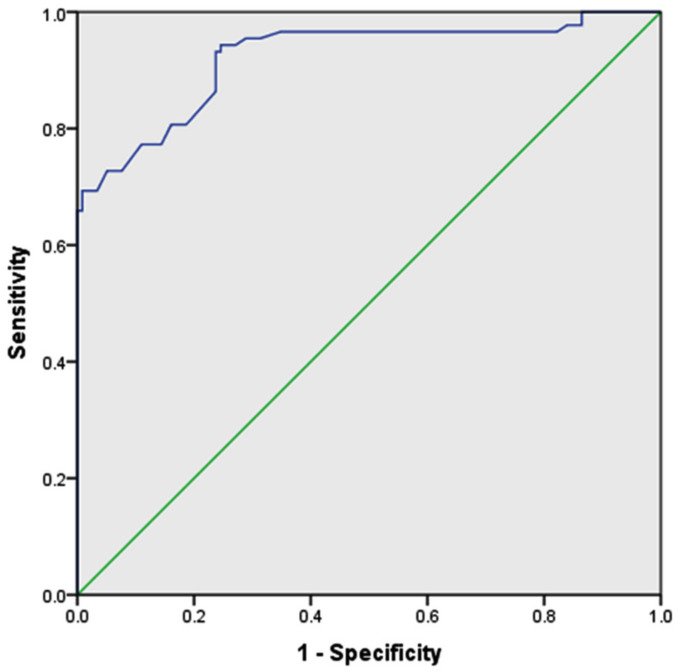
Receiver operating characteristic (ROC) curve for the predictive value of neutrophils count for PCOS. Cut-off value of neutrophil was >46.62 (sensitivity 94.32% and specificity 74.42%) for the prediction of PCOS. The area under the ROC curve (AUC) was 0.922 and Youden index was 0.697 (*p* < 0.05). Diagonal segments were produced by ties.

**Figure 3 medicina-58-00535-f003:**
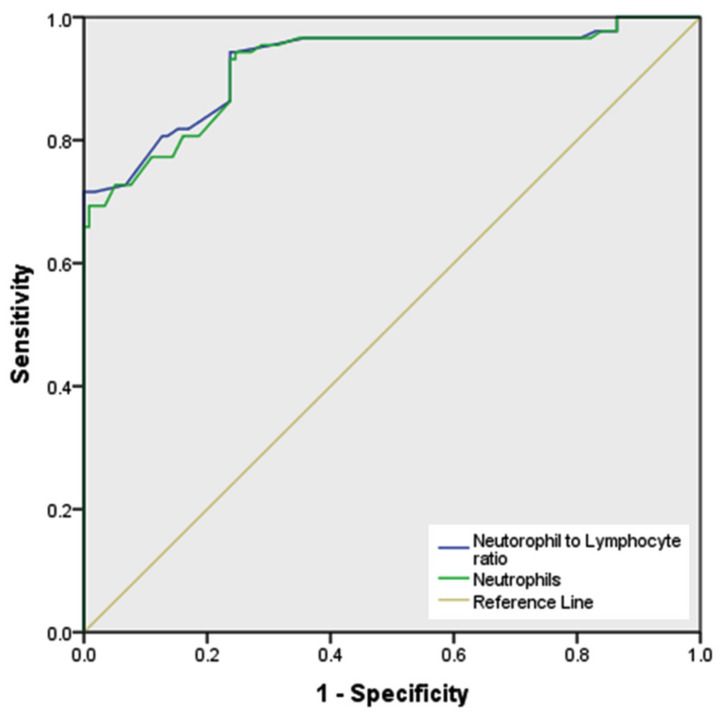
Receiver operating characteristic (ROC) curve for the predictive ability of neutrophil count and neutrophil–lymphocyte ratio (NLR) for the PCOS-associating subclinical inflammation. Cut-off values of neutrophil count and NLR are >46.62 (sensitivity 94.32% and specificity 74.42%), and >1.23 (sensitivity 71.59% and specificity 100%), respectively. Neutrophil area under the curve (AUC) was 0.922 and Youden index was 0.697. AUC for NLR was 0.926 and Youden index was 0.716. Comparative ROC z-statistic value was 2.222, and *p* = 0.026.

**Table 1 medicina-58-00535-t001:** Demographic, anthropometric, and clinical characteristics and laboratory investigations of women with polycystic ovary syndrome (PCOS) compared to healthy age- and BMI-matching women.

Features	PCOS (*n* = 88)	Controls (*n* = 118)	*p* ^a^
Age (years)	21.10 ± 0.30	21.08 ± 0.06	0.740
Body mass index (kg/m^2^)	26.14 ± 1.02	24.43 ± 0.14	**0.000**
Waist circumference (cm)	94.30 ± 0.29	76.04 ± 0.51	**0.000**
Hemoglobin (gm/dL)	12.24 ± 1.10	12.86 ± 0.10	**0.000**
HCT (%)	38.28 ± 3.27	41.81 ± 0.12	**0.000**
Red blood cell count (10^6^/μL)	4.74 ± 0.60	4.13 ± 0.11	**0.000**
MCV (fL)	81.36 ± 11.65	87.48 ± 0.21	**0.000**
MCH (pg)	24.24 ± 0.68	31.09 ± 0.11	**0.000**
MCHC (gm/dL)	31.92 ± 2.33	33.47 ± 0.15	**0.000**
White blood cell count (10^3/^μL)	7.07 ± 2.49	4.87 ± 0.14	0.224
Neutrophils (%)	60.81 ± 10.76	41.44 ± 0.90	**0.000**
Lymphocytes (%)	38.77 ± 10.74	46.99 ± 1.63	**0.000**
Monocytes (%)	0.15 ± 0.38	0.83 ± 0.13	**0.000**
Eosinophils (%)	0.27 ± 0.60	0.41 ± 0.82	0.191
Neutrophil–lymphocyte ratio	1.80 ± 0.95	0.77 ± 0.26	**0.000**
Eosinophil–lymphocyte ratio	0.007 ± 0.01	0.008 ± 0.001	0.738
Platelets–lymphocyte ratio	158.69 ± 109.44	127.33 ± 110.47	**0.000**
Platelets (×10^9^/L)	250.08 ± 87.50	243.87 ± 4.14	0.544
PCT (%)	0.26 ± 0.09	0.60 ± 0.04	**0.000**
MPV (fL)	10.49 ± 1.25	11.18 ± 0.12	**0.000**
Total cholesterol (mg/dL)	183.95 ± 42.58	116.44 ± 1.37	**0.000**
Triglyceride (mg/dL)	143.63 ± 41.93	109.85 ± 1.26	**0.000**
HDL-C (mg/dL)	41.84 ± 6.49	44.05 ± 0.63	**0.000**
LDL-C (mg/dL)	113.78 ± 38.43	47.24 ± 1.88	**0.000**
Total triiodothyronine (ng/mL)	1.56 ± 0.55	1.45 ± 0.06	0.223
Total thyroxine (μg/dL)	8.28 ± 2.47246	7.03 ± 0.12	**0.005**
Thyroid stimulating hormone (μIU/mL)	4.19 ± 1.51	3.99 ± 0.13	0.361
Follicle stimulating hormone (IU/L)	7.38 ± 2.57	8.15 ± 0.13	**0.014**
Luteinizing hormone (IU/L)	14.86 ± 4.75	5.36 ± 0.15	**0.000**
Prolactin (ng/dL)	16.73 ± 6.79	11.94 ± 0.48	**0.000**
Total testosterone (ng/mL)	53.07 ± 28.24	19.23 ± 0.45	**0.000**
Right ovary volume (cc)	17.41 ± 5.99	9 ± 0.00	**0.000**
Left ovary volume (cc)	16.36 ± 4.74	9 ± 0.00	**0.000**

HCT: hematocrit; MCV: mean corpuscular volume; MCH: mean corpuscular hemoglobin; MCHC: mean corpuscular hemoglobin concentration; PCT: Plateletcrit; MPV: mean platelet volume; HDL-C: high-density lipoprotein-cholesterol; LDL-C: low-density lipoprotein-cholesterol. Data are mean ± SD. ^a^ Independent samples *t* test. Statistically significant are in bold

**Table 2 medicina-58-00535-t002:** Pearson’s correlation analysis of neutrophil count vs. the independent laboratory variables investigated in women with polycystic ovary syndrome.

Variable	PCOS	
	*r*	*p*
Age	0.062	0.565
Body mass index	−0.014	0.896
Hemoglobin	0.058	0.591
HCT	0.125	0.248
MCV	−0.135	0.210
MCH	−0.079	0.465
MCHC	−0.013	0.907
White blood cell count	0.182	0.089
Lymphocytes	−0.997	**0.000**
Monocytes	−0.081	0.452
Eosinophils	−0.018	0.865
Neutrophil–lymphocyte ratio	0.930	0.000
Eosinophil–lymphocyte ratio	0.023	0.829
Monocyte–lymphocyte ratio	0.009	0.936
Platelets–lymphocyte ratio	0.674	**0.000**
Platelets	0.139	0.198
Serum cholesterol	−0.095	0.380
Serum triglyceride	0.015	0.887
HDL-C	−0.080	0.458
LDL-C	−0.098	0.363
Total triiodothyronine	0.038	0.729
Total thyroxine	0.050	0.646
Thyroid stimulating hormone	−0.049	0.651
Follicle stimulating hormone	0.042	0.696
Luteinizing hormone	0.196	0.068
Testosterone	0.004	0.969
Right ovary volume	0.150	0.162
Left ovary volume	0.007	0.947
Wrist circumference	0.108	0.318

HCT: hematocrit; MCV: mean corpuscular volume; MCH: mean corpuscular hemoglobin; MCHC: mean corpuscular hemoglobin concentration; HDL-C: high-density lipoprotein-cholesterol; LDL-C: low-density lipoprotein-cholesterol. Data are r and *p* values of 2-tailed Pearson’s correlation. Statistically significant are in bold

**Table 3 medicina-58-00535-t003:** Multi-linear regression analysis for neutrophil count towards polycystic ovary syndrome.

Features	β	*p* Value
Hemoglobin (gm/dL)	0.000	1.000
HCT (%)	0.000	1.000
Red blood cell count (10^6^/μL)	0.000	1.000
MCV (fL)	0.000	1.000
MCH (pg)	0.000	1.000
MCHC (gm/dL)	0.000	1.000
White blood cell count (10^3/^μL)	0.000	1.000
Lymphocytes (%)	**−0.999**	**0.000**
Monocytes (%)	**−0.036**	**0.000**
Eosinophils (%)	**−0.056**	**0.000**
Neutrophil–lymphocyte ratio	0.000	1.000
Eosinophil–lymphocyte ratio	0.000	1.000
Monocyte–lymphocyte ratio	0.000	1.000
Platelets–lymphocyte ratio	0.000	1.000
MPV (fL)	0.000	1.000
PDWC (%)	0.000	1.000
P-LCC (10^3/^μL)	0.000	1.000
P-LCR (%)	0.000	1.000
Total cholesterol (mg/dL)	0.000	1.000
Triglyceride (mg/dL)	0.000	1.000
HDL-C (mg/dL)	0.000	1.000
LDL-C (mg/dL)	0.000	1.000
Total triiodothyronine (ng/mL)	0.000	1.000
Total thyroxine (μg/dL)	0.000	1.000
Thyroid stimulating hormone (μIU/m)	0.000	1.000
Follicle stimulating hormone (IU/L)	0.000	1.000
Luteinizing hormone (IU/L)	0.000	1.000
Total testosterone (ng/mL)	0.000	1.000
Waist circumference (cm)	0.000	1.000
Right ovary volume (cc)	0.000	1.000
Left ovary volume (cc)	0.000	1.000

HCT: hematocrit; MCV: mean corpuscular volume; MCH: mean corpuscular hemoglobin; MCHC: mean corpuscular hemoglobin concentration; MPV: Mean platelet volume; PDWC: platelet distribution width count; P-LCC: Platelet larger cell count; P-LCR; Platelet larger cell ratio; HDL-C: high-density lipoprotein-cholesterol; LDL-C: low-density lipoprotein-cholesterol. Data are β and *p* values of the multi-linear regression analysis. Statistically significant are in bold

## Data Availability

Raw data are available from the corresponding author for interested researchers upon request.

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
