# Peer review of "Quantitative Changes in White Blood Cells: Correlation with the Hallmarks of Polycystic Ovary Syndrome"

_medicina, 2022, doi:10.3390/medicina58040535_

Round 1

Reviewer 1 Report

The authors have shown the relative neutrophilia and elevated NLR are correlated with PCOS. This manuscript is well-organized.  However, correlation is not exactly equal to causation. Thus, it will be more precise to first mention the relationship of parameters of WBC and the metabolic and hormonal biomarkers of PCOS as "correlated". 

A minor comment: Please define WBC and give the full name of "white blood cell" at the beginning of the manuscript. 

Author Response

We sincerely appreciate the time and effort that esteemed reviewers dedicated to evaluate our work . We are grateful for the insightful comments and valuable suggestions made by the reviewers. We have incorporated the suggested recommendations made by the reviewers. For the convenience of the reviewers, the changes made have been highlighted within the manuscript. Please find point by point response to reviewers comments and concerns hereunder.

Comment: The authors have shown the relative neutrophilia and elevated NLR are correlated with PCOS. This manuscript is well-organized.  However, correlation is not exactly equal to causation. Thus, it will be more precise to first mention the relationship of parameters of WBC and the metabolic and hormonal biomarkers of PCOS as "correlated". 

Response:

The same has been taken into account in the revised manuscript.

Minor comment: Please define WBC and give the full name of "white blood cell" at the beginning of the manuscript. 

Needful done.

Reviewer 2 Report

Dear Authors,

This is an interesting study concerning blood parameters on polycystic ovary syndrome.

Here are my observations

  1. Title – please avoid abbreviations
  2. Abstract – please introduce abbreviations when first used in text (e.g. PCOS, WBC)
  3. Abstract (first line) – please use “women of reproductive age” instead of “premenopausal women”
  4. Abstract –please indicate the age ranges since it is important to understand the cardio-metabolic aspects which are more frequent in older adults
  5. Introduction - Rotterdam criteria for teenagers are all 3 elements
  6. Method –exclusion criteria – “congenital adrenal disorders” – did you mean “congenital adrenal hyperplasia”? Did you test 17-hydroxy progesterone since some adult cases might mimic PCOS?
  7. Method – did you exclude Cushing syndrome since this might affect menstruation, inflammation and metabolic complications?
  8. Method – Inflammation might not be related to metabolic syndrome, but with autoimmune or infectious diseases. How did you assess these aspects?
  9. Table 1 – the line with waist circumference should be placed after BMI
  10. Table 1 – please specify total or free testosterone
  11. Table 1– please specify as legend or somewhere in the table the day of menstruation for LH/FSH
  12. Table 1 –how do you explain Total T3 difference among the groups?
  13. Results – did you check high blood pressure?
  14. Did you analyze the subgroup of overweight/obese patients among PCOS?
  15. Did you check other inflammatory markers like PCR, fibrinogen, etc.?
  16. Do you have the data for blood uric acid, as a link between metabolic elements and inflammatory status?

Thank you,

Bests regards,

Author Response

We sincerely appreciate the time and effort that esteemed reviewers dedicated to evaluate our work . We are grateful for the insightful comments and valuable suggestions made by the reviewers. We have incorporated the suggested recommendations made by the reviewers. For the convenience of the reviewers, the changes made have been highlighted within the manuscript. Please find point by point response to reviewers comments and concerns hereunder.

  1. Title – please avoid abbreviations. 

Needful Done 

  1. Abstract – please introduce abbreviations when first used in text (e.g. PCOS, WBC). 

Needful Done. 

  1. Abstract (first line) – please use “women of reproductive age” instead of “premenopausal women”. 

Needful Done 

  1. Abstract –please indicate the age ranges since it is important to understand the cardio-metabolic aspects which are more frequent in older adults. 

The needful is done as per suggestion

  1. Introduction - Rotterdam criteria for teenagers are all 3 elements.

We did not include teenagers 

  1. Method –exclusion criteria – “congenital adrenal disorders” – did you mean “congenital adrenal hyperplasia”? Did you test 17-hydroxy progesterone since some adult cases might mimic PCOS? 

17-hydroxy progesterone was tested for exclusion criteria.

  1. Method – did you exclude Cushing syndrome since this might affect menstruation, inflammation and metabolic complications? 

Yes, indeed Cushing’s syndrome patients were excluded. 

  1. Method – Inflammation might not be related to metabolic syndrome, but with autoimmune or infectious diseases. How did you assess these aspects? 

Our patients have unremarkable medical history other than POCS. Both PCOS and metabolic syndrome are known to be characterized by subclinical inflammation.  

  1. Table 1 – the line with waist circumference should be placed after BMI. 

Needful is Done. 

  1. Table 1 – please specify total or free testosterone. 

We thank the reviewer for pointing this out. Total testosterone was measured and the same has been specified. 

  1. Table 1– please specify as legend or somewhere in the table the day of menstruation for LH/FSH.

Is stated and highlighted in the methods.  

  1. Table 1 –how do you explain the Total T3 difference among the groups? 

It was T4 that may be due to the non-significantly higher TSH and the other pituitary hormones.  

  1. Results – did you check high blood pressure? 

A sub-set of our patients, with metabolic syndrome, were having high blood pressure, but we did not subgroup them accordingly.  

  1. Did you analyze the subgroup of overweight/obese patients among PCOS? 

No., as obese cases were a minority. 

  1. Did you check other inflammatory markers like PCR, fibrinogen, etc.? 

No, we did not check other inflammatory markers.

  1. Do you have the data for blood uric acid, as a link between metabolic elements and inflammatory status? 

No.